# EASY-NET Program: Methods and Preliminary Results of an Audit and Feedback Intervention in the Emergency Care for Acute Myocardial Infarction in the Lazio Region, Italy

**DOI:** 10.3390/healthcare11111651

**Published:** 2023-06-05

**Authors:** Laura Angelici, Carmen Angioletti, Luigi Pinnarelli, Paola Colais, Egidio de Mattia, Nera Agabiti, Marina Davoli, Anna Acampora

**Affiliations:** 1Department of Epidemiology, Regional Health Service–Lazio, Via Cristoforo Colombo, 112, 00147 Rome, Italy; l.pinnarelli@deplazio.it (L.P.); p.colais@deplazio.it (P.C.); n.agabiti@deplazio.it (N.A.); m.davoli@deplazio.it (M.D.); a.acampora@deplazio.it (A.A.); 2Management and Health Laboratory, Institute of Management, Department Embeds, Sant’Anna School of Advanced Studies, 56127 Pisa, Italy; c.angioletti@deplazio.it; 3Critical Pathways and Evaluation Outcome Unit, Fondazione Policlinico Universitario “A. Gemelli”-IRCCS, 00168 Rome, Italy; egidio.demattia@guest.policlinicogemelli.it; 4Faculty of Economics, Università Cattolica del Sacro Cuore, 00168 Rome, Italy

**Keywords:** audit and feedback, emergency, acute myocardial infarction

## Abstract

Within the EASY-NET network program (NET-2016-02364191), Work Package 1 Lazio evaluates the effectiveness of a structured audit and feedback (A&F) intervention compared with the web-based regional periodic publication of indicators in improving the appropriateness and timeliness of emergency healthcare for acute myocardial infarction (AMI). This work describes the A&F methodology and presents the results of the first feedback delivered. The intervention involves sending periodic reports via e-mail to participating hospitals. The feedback reports include a set of volume and quality (process and outcome) indicators, calculated by facility through the health information system of the Lazio Region and compared with regional mean, target values and values calculated for hospitals with similar volumes of activity. Health managers and clinicians of each participating hospital represent the “feedback recipients”. They are invited to organize clinical and organizational audit meetings to identify possible critical issues in the care pathway and define, where necessary, improvement actions. A total of 16 facilities are involved. Twelve facilities present high volumes in all volume indicators, while three facilities present low volumes for each indicator. Concerning the quality indicators, four facilities do not present critical indicators or had average results, three facilities do not present critical indicators but show average results in at least one of the indicators and six facilities present a critical value for at least one of the indicators. The first report highlighted some critical issues in some facilities on several indicators. During the audit meetings, each facility analyzes these issues, defining appropriate improvement actions. The outcome of these actions will be monitored through subsequent reporting to support the continuous care quality improvement process.

## 1. Introduction

In Italy, there is evidence of large variability in healthcare service organization and health outcomes [1]. To reduce this variability and avoid suboptimal implementation of evidence-based practice, various strategies have been proposed, among them audit and feedback (A&F) [2,3]. The WHO defined A&F as “any summary (written or verbal) of clinical performance of health care over a specified period of time” [2]. The purpose of A&F is to measure the performance, compare it to a standard and then to feed the results back to health professionals involved in a particular healthcare pathway with the final goal of improving care. Indeed, A&F is commonly used to help healthcare providers to identify the gap between knowledge and practice and to improve quality of care [3,4]. Providing health professionals with data on their clinical performance should make them aware of the gap to fill [3,4]. According to two Cochrane reviews [5,6], A&F generally leads to small but potentially important improvements in professional practice. Despite A&F as a quality improvement strategy being widely used, evidence suggests that the effects of such interventions vary greatly and are not improving over time [5,6]. Moreover, the optimum methods for implementation and characteristics of A&F strategies that lead to greater impact are still unknown [7,8,9,10]. EASY-NET “Effectiveness of Audit & Feedback strategies to improve healthcare practice and equity in various clinical and organizational settings” is an Italian research network program, articulated in seven Work Packages, co-funded by the Italian Ministry of Health (NET-2016-02364191) and by the involved Italian Regions, including Lazio. The program aims at evaluating the comparative effectiveness of A&F in improving care for different clinical conditions in various organizational and legislative settings and at identifying possible obstacles and facilitating factors to its implementation [11]. Within the program, the Lazio Region, through the Department of Epidemiology of the Regional Health Service (RHS) (namely DEP Lazio), in addition to being the leading Region for the overall program, is developing the Work Package 1 (WP1 Lazio). WP1 Lazio conducts “Comparative Evaluation of Effectiveness of Audit and Feedback (A&F) Strategies to Improve Integrated Care Pathways for Chronic and Acute Conditions” according to two lines of activity, “chronicity”, dedicated to healthcare for chronic conditions, and “emergency”, dedicated to healthcare for acute conditions [12]. The research activities within the WP1-Lazio Emergency aim to compare effectiveness in improving the appropriateness and timeliness of emergency health interventions for acute myocardial infarction (AMI) between a structured A&F strategy and the periodic web-based publication of indicators (“standard strategy”). DEP Lazio annually calculates and publishes results of many process and outcome indicators, evaluating healthcare for different health conditions (both acute and chronic) provided either in hospital or in community settings, through a dedicated regional web platform called P.Re.Val.E (“Programma Regionale Valutazione Esiti”—Regional Program for Outcomes and Processes Evaluation) [13]. This platform is publicly accessible, and its annual update is promoted in an open meeting to which healthcare management professionals of all the healthcare facilities of the region are invited. Using a specific function available through the platform, a healthcare provider has the possibility to activate an Audit procedure also involving DEP Lazio, but this activity is on the providers’ initiative. Therefore, in this “standard strategy”, the “feedback” is returned to providers through a web publication, and no other initiatives are offered by DEP Lazio. A structured A&F intervention has been defined within the WP1 Lazio Emergency project, taking into account the most recent evidence in the field suggesting ways of optimizing these strategies [9,14,15]. 

The objectives of this work were to describe the A&F methodology for improving the quality of the in-hospital emergency care pathway for AMI implemented in the Lazio region and to report the results of the first delivered feedback.

## 2. Materials and Methods

### 2.1. Study Design and Participants

All fifty hospitals in the Lazio region were invited to participate to a prospective quasi-experimental, pre–post study with a control group, aimed at evaluating the effectiveness of an A&F intervention in promoting quality of in-hospital emergency care for patients affected by AMI and centrally coordinated by DEP Lazio. A project team (PT) was defined, consisting of experienced methodologists from DEP Lazio, experienced clinicians (cardiology and emergency medicine) and healthcare management professionals from involved hospitals. Although the intervention focused on the in-hospital AMI emergency care pathway, the regional emergency medical service (ARES 118 “Agenzia Regionale Emergenza Sanitaria”) and the Regional Health and Social-Health Integration Directorate—Hospital and Specialty Network Area of the Lazio Region were also invited to collaborate, thus accounting also for the overall pathway.

### 2.2. A&F Intervention

The experimental A&F intervention started in December 2021 and is scheduled to finish in September 2023. As the first step, a kickoff meeting was organized to present the project activities and to collect agreement to participate from healthcare managers and clinicians—“feedback recipients”—of all fifty hospitals in the Lazio region. Before the intervention started, a questionnaire was administered to collect information about the state of implementation of A&F and assimilating activities in the participating hospital. Methods and results of this survey are described in the study by Angioletti et al. [16].

The A&F intervention includes:-Sending periodic reports (feedback) with the results of a set of process and outcome indicators evaluating the in-hospital emergency care for AMI patients, in a defined reference period;-Inviting the recipients to organize audit meetings in which they should discuss the reported results, identify critical issues and define improvement actions;-Inviting the recipients to return a form collecting information about the audit meetings characteristics to the research team.

The main features for drafting the feedback reports were defined in collaboration with the PT and by administering a questionnaire to all the recipients for selecting the graphs, tables and textual information they prefer. The overall intervention, lasting two years, involves the delivery of feedback four times, once every six months. By the end of February of each year, recipients are provided with preliminary feedback reporting the results related to the first six months of the previous year, and by the end of September they receive consolidated feedback reporting data related to the entire previous year.

Feedback is delivered both in verbal and written form. First, participants are invited to an in-person meeting to introduce the feedback, to receive their “feedback on the feedback” using structured questionnaires and to collect their experience.

The written feedback is delivered in two forms. A main document reports the results of the set of indicators calculated for all participating hospitals. Results are reported using bar graphs in which each bar corresponds to a specific hospital and one or more dotted lines display reference values. Reference values may be “standard values” from current national or regional regulation (if available) and/or the values calculated at regional level (Lazio) as reported in P.Re.Val.E. Results are summarized above the figure. The second provided document is a hospital-specific slide presentation comparing hospital data with the mean value of all hospitals in Lazio region with similar volume of activities. This value is used as reference since it is demonstrated that a higher volume of activity is associated with better outcomes.

After receiving feedback, recipients are invited to organize clinical and organizational audit meetings in their respective facilities with the aim to identify possible critical issues and to define, where necessary, improvement actions, timing for actions and to nominate a local responsible for the implementation. An “audit form” is provided, along with the feedback, to collect information about the audit’s characteristics and also as an organizational guide. The information collected is reported in Appendix A. During the “inter-feedback” period, recipients have the possibility to contact DEP Lazio (the feedback provider) for any support request. Furthermore, the research team regularly contacts recipients by phone and/or by e-mail with the purpose of updating the activities and collecting any issues or suggestions.

### 2.3. Indicators and Population

Indicators included in the feedback were selected, in collaboration with health professionals participating in the PT, by using a modified Delphi method, i.e., a group process used to survey and collect the opinions of experts on a particular subject [15,17].

The preliminary list of indicators evaluated for inclusion comprised some of those regularly published by the P.Re.Val.E. program [13] and additional ones proposed by TP members. Table 1 shows the nine indicators reported in the first feedback.

The list of indicators includes:-Three volume indicators to provide a description of the amount of activity in each participating facility, which are useful for the interpretation of process and outcome indicators.

Two reference values for comparison are defined according to evidence and Italian legislation. The decree of the specially appointed commissioner of the Lazio Region (“Decreto del Commissario ad Acta” DCA) “DCA 412/2014” defined a minimum annual volume of 300 hospitalizations for AMI and at least 400 percutaneous transluminal coronary angioplasties (PTCAs). At national level, the Ministry Decree (“Decreto Ministeriale-DM”) “DM 70/2015” [18] indicated a minimum annual volume of 100 hospitalizations for AMI and at least 250 PTCAs.

-Two process indicators, evaluating PTCA in ST-segment-elevation myocardial infarction (STEMI) patients. In Italy, the DM 70/2015 indicates as a target value a minimum proportion of patients treated with PTCA within 90 min of 60%. The second indicator is a modified version of the first one calculated with the intention of excluding those patients for whom PTCA is no longer appropriate due to the large timeframe (longer than 12 h) passed since the access in the ER. In these patients, we can consider that PTCA was appropriately not performed.-Four outcome indicators, evaluating 30-day mortality and in-hospital mortality for AMI and STEMI inpatients. As no standard values are available for these indicators, the regional mean was used as reference value.

In addition, all indicators calculated by facility were compared to values calculated for groups of hospitals characterized by similar volumes of activity.

In the final part of the feedback report, two summary grids are provided for an overall view of all the indicators for all participating hospitals. The grids are defined based on a series of cut-off points starting from the target values indicated by the literature and by current legislation (where available). According to the P.Re.Val.E., previously identified categories were used [13].

Each indicator for each hospital is colored according to a color scale proportionally to the corresponding score. The synthetic grid for the volume indicators reports each class of volume according to a scale of blue where the dark blue corresponds to the highest volume (volume < 5 are indicated in gray). The synthetic grid for the process and outcome indicators reports each class of performance according to a red-to-green scale where the dark red corresponds to the lowest performance and the dark green to the highest one (hospitals with volume of activity < 5 are indicated in gray). For comparison purposes, the second grid is populated based on adjusted measures. Where it was not possible to estimate adjusted measures, the crude values are reported and identified by a star (*).

Reading the grid along the row, it is possible to visually evaluate the overall performance of a hospital and to identify critical indicators. On the contrary, reading the column, it is possible to compare the performance measured by a particular indicator across all the participating hospitals and to identify hospitals for which that indicator is critical.

### 2.4. Data Sources

The indicators were calculated using pseudo-anonymized data collected through the health information system (HIS) of the Lazio Region. Specifically, the data sources used include the Italian Hospital Discharge Registry (HDR), the Healthcare Emergency Information System (HEIS) and the Tax Registry. The HDR information system contains sociodemographic and clinical data systematically collected during each hospital admission and discharge from facilities of the Lazio region, including the main and additional diagnoses and all the procedures carried out. Eligibility and exclusion criteria for the selection of the cohort of interest were defined according to the International Classification of Disease, Ninth Revision, Clinical Modification, (ICD-9-CM) codes [19]. Codes for each indicator are reported in Appendix A.

An anonymous identification code, assigned from the HIS, was used as the key for the record-linkage procedure applied using the deterministic methodology. Data from the HDR were linked to data collected through the Information System of Health Emergency (HEIS) that routinely collects sociodemographic and clinical information about treatments and access to all the Emergency Department of hospitals in the Lazio region and the “Tax Registry” that includes information on deaths.

By integrating data from different data sources, a demographic and health-related profile is defined, allowing for patients’ clinical history for the five years preceding the admission of interest to be traced.

### 2.5. Data Management and Statistical Analysis

The calculation formulas of the included indicators are reported in Appendix A. Each indicator is calculated by facility for groups of hospitals with the same volume of activity and at the regional level [13,20,21].

The adjusted measures and related 95% CIs are calculated by generalized linear models with binomial distribution and logit as a link function adjusting for demographics and clinical characteristics selected by means of a stepwise procedure (Appendix A). The threshold values used to define the classes of performance reported in the grids were defined using the “Jenks natural breaks” algorithm [22]. These values are dynamic, and the identification of classes needs to be periodically updated. Statistical analyses were performed using SAS version 9.2 (SAS Institute Inc., Cary, NC, USA).

## 3. Results

Out of the total of 43 facilities in the cardiological emergency network in Lazio region, 16 participate in the intervention for the AMI pathway evaluation. We report a selection of the results from the first feedback for the most relevant volume and quality indicators. All other results are available in Appendix A.

The results presented refer to the first reporting period, namely 1 January to the 31 December 2021. The feedback was delivered by the end of September 2022.

### 3.1. Volume Indicators

In 2021, 7766 hospitalizations for AMI were reported in the Lazio region, of which 3249 (41.8%) were STEMI. Hospitalizations for AMI have been progressively decreasing since 2015 and for STEMI since 2010 [23]. Hospitalizations for both conditions fell dramatically in 2020 as compared to 2019, greater for AMI (−17.7%) than for STEMI (−12%), largely due to the COVID-19 pandemic (Table 2). The progressive decline in hospitalizations for AMI and STEMI is in line with national and international evidence [23].

Figure 1 shows the volume of activities for AMI by participating facilities in 2021. Considering results calculated for the entire year 2021, 6 of the 16 facilities (38%) met the target defined by DCA 412/2014. Twelve facilities (75%) met the target defined by DM 70/2015 (details for years 2019, 2020 and 2021 are given in Table 2). Since 2020, a general reduction in hospitalization for AMI is shown.

Figure 2 shows the activity volumes per facility in 2021, relating only to STEMI hospitalizations in which at least one PTCA was performed. In 2021, 4 out of 16 facilities (25%) reported more than 150 STEMI admissions with at least one PTCA and 5 out of 16 (31%) between 100 and 150 admissions per year. One facility has less than 50 admissions per year, while four facilities reported no STEMI admissions with at least one PTCA (data for years 2019, 2020 and 2021 are given in Table 3).

### 3.2. Process Indicators

Figure 3 shows the proportion of PTCAs performed within 90 min from the access to emergency room (ER) in patients diagnosed with STEMI, by facility in 2021. Five out of sixteen facilities (31.3%) meet the minimum target value of 60%. Further three facilities are slightly below the minimum target value. The remaining three facilities are significantly below the minimum target value (raw and adjusted proportions are reported in Table 4). Finally, five facilities had very low values, which did not permit calculation of the adjusted indicator (raw proportions in Table 4).

Considering the volumes of PTCA performed in STEMI patients for the year 2021 (Figure 3 and Table 4), facilities were divided into four volume categories (≤50; 51–100; 101–150; >150 number of PTCA). Each facility can then be compared with the value calculated for all the facilities of the Lazio region in the same category (also not participants). The graphs by facilities are shown in Appendix A. In general, five facilities present lower adjusted proportions than the value calculated within the same volume class.

### 3.3. Outcome Indicators

Figure 4 shows the mortality within 30 days from the first hospital admission of patients diagnosed with STEMI, by facility. Results for the year 2021 are shown in Figure 4 and Table 5. No standard reference value is available for this indicator. Four facilities have showed values above the regional reference value. Three facilities had very low values, which made it impossible to calculate the adjusted mortality. The crude proportion is reported and is higher than the regional value in all cases. For one facility, on the other hand, the crude mortality is zero (crude and adjusted mortality reported in Table 5).

Considering the admission volumes for STEMI for the year 2021 (Appendix A), the facilities were divided into four volume categories (≤50; 51–100; 101–150; >150 number of PTCA). Each facility can then be compared with the value calculated for all the facilities of the Lazio region in the same category (also not participants). The graphs by facilities are shown in Appendix A. In general, six structures show a higher adjusted mortality than the value calculated within the same volume class.

Figure 5A,B summarize the results of the indicators calculated by facility. Twelve facilities present high or very high (AMI and STEMI > 51) volumes in all volume indicators, while three facilities present low or very low volumes (AMI and STEMI ≤ 50). Concerning the quality indicators, four facilities do not present critical indicators or have average results (very high or high), three facilities do not present critical indicators but average results in at least one of the indicators (very high or high or medium), six facilities present for at least one of the indicators a critical or very critical value (low or very low) and for two facilities it is not possible to calculate the adjusted value for any of the included indicators.

## 4. Discussion

The objectives of the present work were to describe a new audit and feedback intervention for improving the quality of the in-hospital emergency care pathway for AMI implemented in the Lazio region and to report the results of the first delivered feedback. This research is part of the EASY-NET research program (NET-2016-02364191), and the final goal is to compare the efficacy of the intervention in improving timeliness and outcome of emergency care for AMI as compared to the existing A&F strategy in the region [13]. The study involved all hospitals in Lazio; those that adhered to the intervention are considered as “exposed” and those that did not adhere as “not exposed”. In order to account for previously existing A&F strategies at the hospital level, we performed a baseline analysis of the state of the art. The study by Angioletti et al. highlighted that A&F, and assimilating activities, are widely used among the hospitals of the Lazio region, but there is a large variability in their characteristics. In most cases, these are limited to the discussion of clinical cases or the consultation of indicators, while the definition of improving activities and the identification of those responsible for the implementation of these activities is rarely reported [16]. Starting from this baseline picture, we elaborated an A&F strategy consisting mainly of the delivery of periodic feedback reports, the invitation to organize audit meeting starting from the reported results and the invitation to periodic in-person meetings with the feedback provider (DEP Lazio). The strategy was designed taking into account the most recent evidence on the characteristics that could improve the effectiveness of the feedback. Recently, Brehaut and colleagues [9] identified 15 key suggestions for designing and delivering effective practice feedback interventions. Although not all suggestions could be applied to all A&F strategies, they represent a useful guide to design an A&F intervention informed by the evidence. According to the first group of suggestions, regarding the type of actions that recipients should put into practice to improve their performance, we focused on the in-hospital part of the emergency care pathway for AMI by setting the specific goal to improve the timeliness of care (e.g., PTCA) that is essential to reach better outcomes (e.g., reducing mortality for AMI). In this context, the improving actions are specific and under the control of the recipients. After receiving the feedback, recipients have the opportunity, if relevant, to ask the provider (DEP Lazio) for detailed information at patient level, allowing re-examination of the cases to better understand how they can improve. Secondly, as the authors suggested, we planned to deliver feedback periodically every six months and to provide data at unit level (a small group of different professionals from different disciplines working together). In order to optimize the feedback display, we included synthetic written messages along with graphical data representation. The intervention was planned in collaboration with the recipients. The indicators used for evaluating the care pathway of interest and the main features of the feedback report (graph, tables, text) were defined by consulting participating health professionals (recipients) using a structured methodology (modified Delphi methods [15,17] and structured electronic surveys). In particular, an objective and subjective evaluation of the clarity of different graphical solutions was performed by conducting a survey involving the recipients. For selecting the graphics to use, the objective evaluation (correct interpretation assessed through specific questions) took priority over the subjective evaluation (collected as personal preference). The active involvement in constructing the feedback, along with the use of a limited but representative group of indicators, made it possible to reduce the cognitive load for the recipients when they receive the feedback. Finally, during the face-to-face meetings, we periodically collected the “feedback on the feedback” from the recipients, with the objective of improving the materials provided. Indeed, it has been reported in literature that socially constructed feedback could be more effective [9,14].

On the other hand, it must be considered that feedback delivered by a research team, as in our case, might be less effective than that delivered by a supervisor or a respected colleague [9]. This could be due to the scarce credibility to the feedback process (i.e., available data, data collection process, data analysis) [9]. To overcome this possible limitation, we made all efforts to address the credibility of the provided information. For example, participants have the opportunity to ask any questions or for clarifications both during the in-person meeting and via email, call/videocall or online meeting.

A total of 16 health facilities of the Lazio region participate in the EASY-NET program for the AMI pathway evaluation. The indicators selected included volume, timeliness and outcome indicators. The volume of activities represents a measurable process and can serve as effect modifiers with respect to the effectiveness of healthcare interventions. There is a great body of literature that demonstrates the association between the volume of activities and outcomes of health interventions [24]. Hospitalizations for AMI and PTCA are among the interventions for which the association between the volume of activities and the patient outcomes (in-hospital and 30-day mortality) has been demonstrated [21].

In Italy, target volumes are currently defined by the DM 70/2015 and, at a regional level, by the DCA 412/2014. DM 70/2015 defines target volumes corresponding to at least 100 admissions for AMI and at least 250 PTCAs in a year. DCA 412/2014 sets targets for a volume of at least 300 admissions for AMIs and at least 400 PTCAs.

Six of the sixteen hospitals (38%) met the target values defined by the DCA 412/2014, and twelve (75%) met the target values defined by DM 70/2015. Furthermore, 4 of 16 reported (25%) more than 150 STEMI admissions with at least one PTCA and 5 (31%) between 100 and 150 admissions per year. One facility has less than 50 admissions per year, while four facilities reported no STEMI admissions with at least one PTCA.

Regarding the process indicators, is well documented that timely and effective treatments are essential for the survival of the patient with AMI [25]. In recent years, 30 day-mortality from AMI has been significantly reduced [26]. In the Lazio region, according to P.Re.Val.E. program data, over the last 10 years, the 30-day mortality for AMI has also been reduced from 9.69% (95% CI: 9.08–10.30%) to 7.55% (95% CI: 6.93–8.17%), while the percentage of patients with STEMI who promptly underwent PTCA has increased from 29.54% (95% CI: 28.1–30.98%) to 55.48% (95% CI: 53.56–57.40%) [13,23].

Considering the proportion of PTCAs performed within 90 min from the first access to the ER in patients diagnosed with STEMI, only 5 out of 16 facilities (31.3%) met the minimum target value of 60% in 2021, while three presented values significantly under the target. In those cases, a recommendation to carry out a revision of all the phases of the in-hospital pathway was formulated.

Finally, the outcome indicators included the 30-day mortality (after discharge), an indicator that reflects at least in part the quality of care provided to the patient. In the Lazio region, this appears to be declining in recent years for both AMI and STEMI from 10.6% to 7.1% and from 12.3% to 9% from 2010 to 2019, respectively. In 2020, again in relation to the COVID-19 pandemic, a reversal is observed with a slight increase to 7.8% and 9.6%, respectively [13,23].

Regarding the participating hospitals, four had an adjusted 30-day mortality in STEMI patients above the regional value, and six showed a higher adjusted mortality than the value calculated for hospitals within the same volume class. Although the 30-day mortality depends only in part on the in-hospital care, professionals were invited to revise the clinical cases of AMI patients who died during the 30 days after the discharge with the aim to identify possible critical points also considering the in-hospital mortality.

This research presents different strengths as well as limitations.

Firstly, the described methodology was designed considering the most relevant and recent evidence on the ways of optimizing A&F, including engagement of participants in the development phase. A second strength is related to the data sources used that are represented by the HIS that represent a very comprehensive source of information. Furthermore, since they are regularly collected, they provide the opportunity for healthcare organizations and systems to develop regular quality improvement programs. Beyond the strengths, they also present limitations, mainly related to the quality of the collected data [27,28]. As the process of entering incorrect data causes erroneous results [29], it is of paramount importance to audit the quality of the data recorded in the health information systems in order to ensure the correct handling of the input data [30] involved in the calculation of the indicators. Moreover, the research team produces and sends well defined and standardized feedback reports, thus giving hospitals the possibility to organize the audit meetings independently. This can certainly create problems of heterogeneity and coordination among the participating facilities but gives them the possibility to adapt the meetings to their own needs. In any case, to improve coordination, periodic meetings for discussion were organized, and numerous contacts were held via email and telephone with all the participants.

During the experimental intervention, after receiving the feedback, participants (“exposed”) could observe a discrepancy between the provided data and their performance. In these cases, they were invited to require data quality audits involving the feedback providers. The P.Re.Val.E. program [13,23] (available for all the hospitals in the region, including the “not exposed”) regularly conducts data quality audits in hospitals with results which deviate significantly from the regional average. The aim of these audits is to identify possible critical points, which can require specific interventions to improve data quality.

## 5. Conclusions

Although the difference in the effect measurements across studies is large, A&F is recognized as an effective strategy. The effectiveness of these interventions depends on how it is constructed and delivered. Our analysis of hospital performance highlighted heterogeneity among participating hospitals and some critical issues in some of them on a variable number of indicators. The A&F methodology designed and implemented in the EASY-NET project in the Lazio region represents a useful instrument to promote quality of care. The following feedback reports, delivered to the recipients, will support clinical-organizational audit meetings, as each facility will be able to analyze critical issues, involving all relevant figures and define appropriate improvement actions. The outcome of these actions will be monitored through subsequent reporting to support the continuous care quality improvement process. Improvements will be monitored during the entire study period and, finally, analyzed to demonstrate the efficacy of the experimented intervention.

## Figures and Tables

**Figure 1 healthcare-11-01651-f001:**
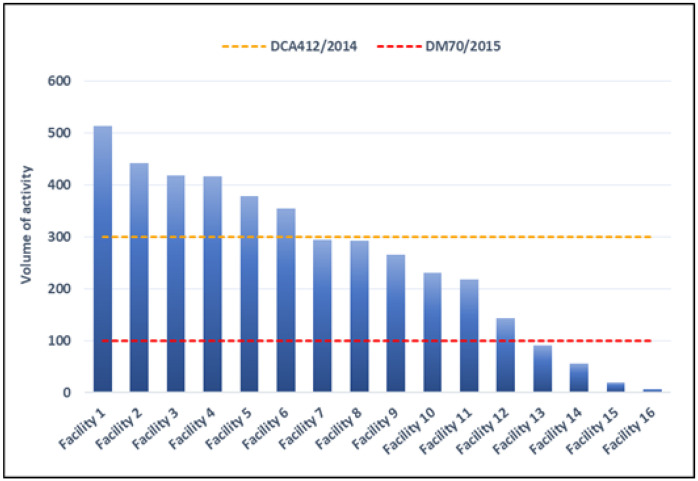
Number of hospitalizations of patients with AMI by facility (2021).

**Figure 2 healthcare-11-01651-f002:**
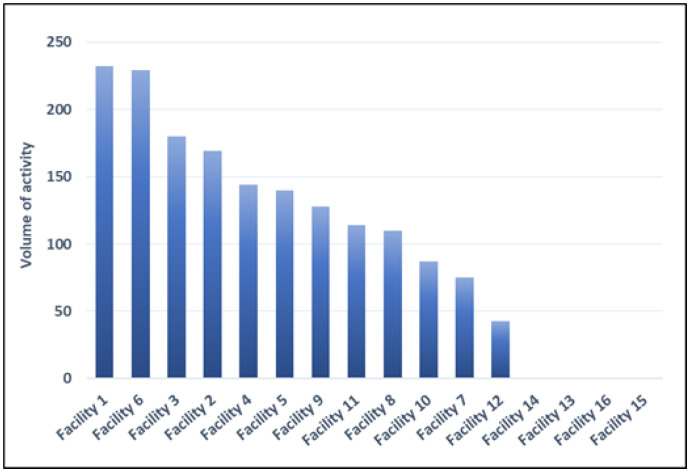
Number of hospitalizations of patients with STEMI treated with PTCA by facility (2021).

**Figure 3 healthcare-11-01651-f003:**
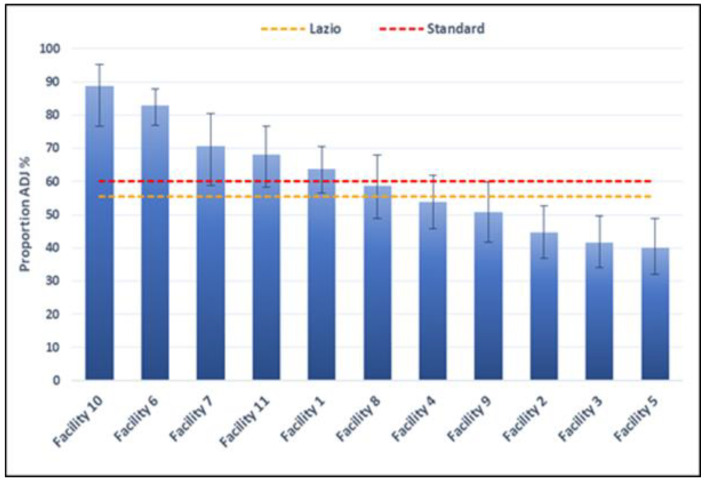
Proportion of patients with STEMI treated with PTCA within 90 min from access to ER, by facility (2021).

**Figure 4 healthcare-11-01651-f004:**
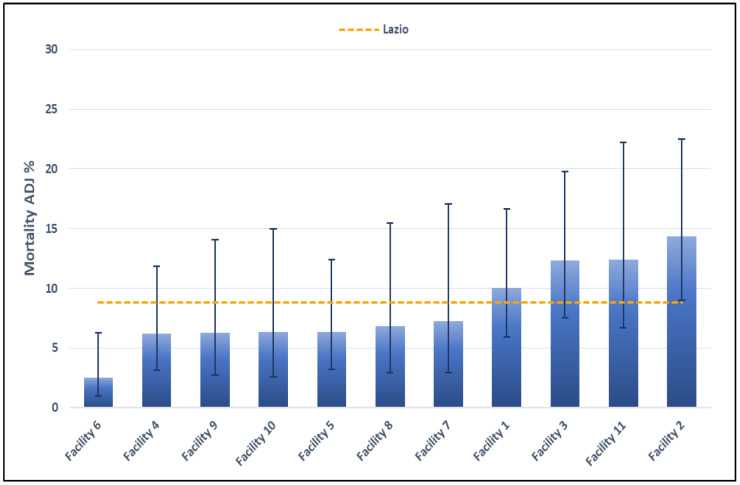
Mortality within 30 days after first admission to hospital for STEMI, by facility (2021).

**Figure 5 healthcare-11-01651-f005:**
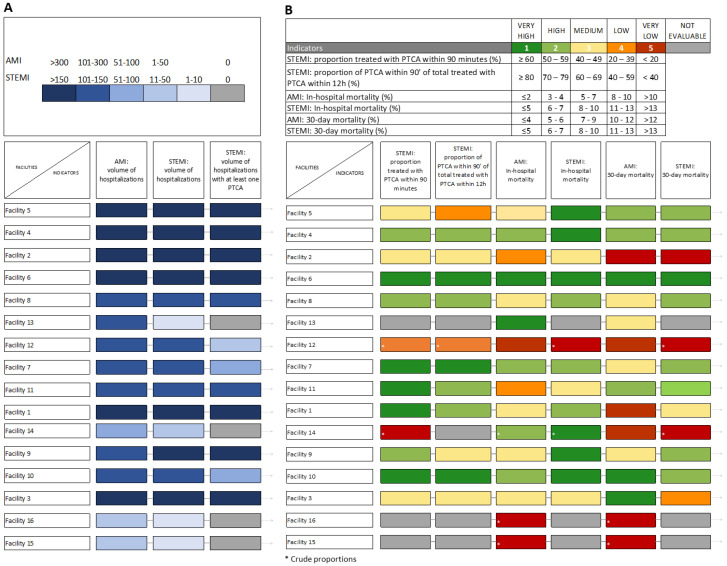
Summary grid for (**A**) volume indicators (2021) and (**B**) process and outcome indicators (ADJ % 2021).

**Table 1 healthcare-11-01651-t001:** Volume, process and outcome indicators for the evaluation of in-hospital emergency care for AMI patients.

Indicators	Calculation	Reference Value
AMI: volume of hospitalizations	Number of hospitalizations of patients diagnosed with AMI	100/y (DM 70/2015)
STEMI: volume of hospitalizations	Number of hospitalizations of patients diagnosed with STEMI	NA
STEMI: volume of hospitalizations with at least one PTCA	Number of hospitalizations of patients diagnosed with STEMI who received at least one PTCA	NA
STEMI: proportion treated with PTCA within 90 min	Number of hospitalized patients diagnosed with STEMI who received PTCA within 90 min from access to the ER/number of hospitalization of patients diagnosed with STEMI	≥60% (DM 70/2015)
STEMI: proportion of PTCA within 90 min of those treated with PTCA within 12 h	Number of hospitalized patients diagnosed with STEMI who received PTCA within 90 min from access to the ER/number of hospitalization of patients diagnosed with STEMI who received PTCA within 12 h	NA Compared to regional mean
AMI: in-hospital mortality	Number of hospitalized patients diagnosed with AMI who died during the hospital stay/number of hospitalized patients diagnosed with AMI	NA Compared to regional mean
STEMI: in-hospital mortality	Number of hospitalized patients diagnosed with STEMI who died during the hospital stay/number of hospitalized patients diagnosed with STEMI	NA Compared to regional mean
AMI: 30-day mortality	Number of hospitalized patients diagnosed with STEMI who died during the 30 days after hospital discharge/number of hospitalized patients diagnosed with AMI	NA Compared to regional mean
STEMI: 30-day mortality	Number hospitalized patients diagnosed with STEMI who died during the 30 days after hospital discharge/number of hospitalized patients diagnosed with STEMI	NA Compared to regional mean

STEMI: ST-elevation myocardial infarction, PTCA: percutaneous transluminal coronary angioplasty, ER: emergency room, NA: not available.

**Table 2 healthcare-11-01651-t002:** Number of hospitalizations of patients with AMI by facility (2019, 2020, 2021).

Facility	Volume of Activity		
	2019	2020	2021
Lazio	9794	8061	7766
Facility 1	710	584	514
Facility 2	506	413	442
Facility 3	696	420	418
Facility 4	470	363	416
Facility 5	491	427	378
Facility 6	426	378	354
Facility 7	399	306	295
Facility 8	289	266	292
Facility 9	389	306	266
Facility 10	396	323	231
Facility 11	253	230	218
Facility 12	305	237	143
Facility 13	97	75	91
Facility 14	106	72	56
Facility 15	51	27	20
Facility 16	9	10	7

**Table 3 healthcare-11-01651-t003:** Number of hospitalizations of patients with STEMI treated with PTCA by facility (2019, 2020, 2021).

Facility	Volume of Activity		
	2019	2020	2021
Lazio	2870	2625	2719
Facility 1	369	274	232
Facility 6	215	188	229
Facility 3	217	178	180
Facility 2	172	140	169
Facility 4	162	139	144
Facility 5	156	146	140
Facility 9	172	118	128
Facility 11	121	121	114
Facility 8	114	113	110
Facility 10	112	99	87
Facility 7	87	80	75
Facility 12	105	72	43
Facility 14	0	0	0
Facility 13	0	1	0
Facility 16	1	0	0
Facility 15	0	0	0

**Table 4 healthcare-11-01651-t004:** Proportion of patients with STEMI treated with PTCA within 90 min from access to ER, by facility (2021).

Facility	N	Crude %	95% CI		Adj %	95% CI	
Lazio	2601	55.48	53.56	57.38	_	_	_
Facility 10	55	89.09	78.17	94.90	88.80	76.56	95.25
Facility 6	206	82.04	76.23	86.68	82.98	76.89	87.77
Facility 7	71	67.61	56.06	77.34	70.66	58.68	80.39
Facility 11	109	67.89	58.64	75.92	68.15	58.37	76.60
Facility 1	201	66.67	59.89	72.82	63.85	56.60	70.53
Facility 8	110	61.82	52.49	70.35	58.73	48.91	67.93
Facility 4	164	51.83	44.23	59.35	53.93	45.84	61.84
Facility 9	122	54.10	45.27	62.68	50.76	41.70	59.79
Facility 2	161	45.96	38.45	53.67	44.68	36.97	52.67
Facility 3	168	40.48	33.35	48.03	41.62	34.09	49.56
Facility 5	136	40.44	32.57	48.84	40.14	32.00	48.88
Facility 12	39	38.46	24.89	54.10	_	_	_
Facility 14	6	0.00	0.00	_	_	_	_
Facility 15	3	0.00	0.00	_	_	_	_
Facility 13	2	0.00	0.00	_	_	_	_
Facility 16	1	0.00	0.00	_	_	_	_

**Table 5 healthcare-11-01651-t005:** Mortality within 30 days after first admission to hospital for STEMI, by facility (2021).

Facility	N	Crude %	95% CI		Adj %	95% CI	
Lazio	2623	8.84	7.82	9.99	_	_	_
Facility 6	206	2.43	1.04	5.56	2.54	1.01	6.31
Facility 4	164	7.32	4.23	12.35	6.18	3.17	11.86
Facility 9	122	4.92	2.27	10.32	6.31	2.75	14.09
Facility 10	77	7.79	3.62	15.98	6.36	2.61	14.98
Facility 5	136	8.09	4.58	13.90	6.38	3.22	12.40
Facility 8	110	5.45	2.52	11.39	6.82	2.92	15.45
Facility 7	71	8.45	3.93	17.24	7.24	2.96	17.08
Facility 1	201	8.46	5.35	13.13	10.01	5.90	16.68
Facility 3	168	11.90	7.84	17.67	12.32	7.53	19.79
Facility 11	109	11.93	7.10	19.34	12.42	6.73	22.23
Facility 2	161	13.66	9.20	19.82	14.38	8.99	22.53
Facility 12	39	25.64	14.57	41.08	_	_	_
Facility 15	3	33.33	6.15	79.23	_	_	_
Facility 14	6	33.33	9.68	70.00	_	_	_
Facility 13	2	0.00	_	_	_	_	_
Facility 16	1	0.00	_	_	_	_	_

## Data Availability

Data related to the findings reported in our manuscript are available to all interested researchers upon reasonable request and with the permission of the Regional Department because of stringent legal restrictions regarding the privacy policy on personal information in Italy (national legislative decree on privacy policy n. 196/30 June 2003). For these reasons, our dataset cannot be made available on a public data repository.

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
