# Peer review of "EASY-NET Program: Methods and Preliminary Results of an Audit and Feedback Intervention in the Emergency Care for Acute Myocardial Infarction in the Lazio Region, Italy"

_healthcare, 2023, doi:10.3390/healthcare11111651_

Round 1
Reviewer 1 Report
The problem addressed is particularly interesting.
The description of a new Audit &Feedback Intervention could improve the quality of the medical act and implicitly the prognosis of patients with Acute Myocardial Infarction.
Audit &Feedback Intervention methodology designed and implemented in EASY-NET project can represent a useful instrument to promote quality of care not only for acute myocardial infarction patients but also in cases of other medical emergency.
The methodology of the study is well described, the discussions are well argued, and the conclusions are pertinent.
these are my supplementary comments:
1) The study analyzed the differences in the application of the guidelines and the optimal treatment in the case of patients with acute myocardial infarction, one of the main causes of mortality and morbidity (in the present case in Italy).
2) Treatment differences were analyzed within a large number of hospitals (n = 16).
3) Current data were presented regarding some indicators such as: volume of hospitalization (AMI and STEMI), proportion of STEMI patients treated with PTCA within 90 minutes (or 12 hours) - thus creating some contemporary reference values in Europe.
4)In the current study were raised problems of adherence to the management guidelines of patients with AMI in Europe, but also solutions for solving them (which can be extrapolated to other hospitals in Europe or other geographical areas).
5) The authors also presented data on mortality in a contemporary cohort of patients with AMI, from a large number of medical centers, being a reference and starting point for another region and hospitals.
6) Solutions were analyzed to make the treatment of STEMI patients more efficient, so that all patients in all medical centers benefit from quality treatment (which can also be applied to other countries to increase adherence to the guidelines).
Congratulations to the authors!
Author Response
Dear Reviewer,
We warmly thank the reviewers for their comments, and we took the opportunity of this revision to carefully review the English and also to revise the formatting of the article. All the changes made are reflected in the revised version of the manuscript in yellow highlight.
Response to Reviewer 1 Comments
The problem addressed is particularly interesting. The description of a new Audit &Feedback Intervention could improve the quality of the medical act and implicitly the prognosis of patients with Acute Myocardial Infarction. Audit &Feedback Intervention methodology designed and implemented in EASY-NET project can represent a useful instrument to promote quality of care not only for acute myocardial infarction patients but also in cases of other medical emergency. The methodology of the study is well described, the discussions are well argued, and the conclusions are pertinent.
These are my supplementary comments:
- The study analyzed the differences in the application of the guidelines and the optimal treatment in the case of patients with acute myocardial infarction, one of the main causes of mortality and morbidity (in the present case in Italy).
- Treatment differences were analyzed within a large number of hospitals (n = 16).
- Current data were presented regarding some indicators such as: volume of hospitalization (AMI and STEMI), proportion of STEMI patients treated with PTCA within 90 minutes (or 12 hours) - thus creating some contemporary reference values in Europe.
- In the current study were raised problems of adherence to the management guidelines of patients with AMI in Europe, but also solutions for solving them (which can be extrapolated to other hospitals in Europe or other geographical areas).
- The authors also presented data on mortality in a contemporary cohort of patients with AMI, from a large number of medical centers, being a reference and starting point for another region and hospitals.
- Solutions were analyzed to make the treatment of STEMI patients more efficient, so that all patients in all medical centers benefit from quality treatment (which can also be applied to other countries to increase adherence to the guidelines).
Congratulations to the authors!
We thank Reviewer 1 for effectively summarizing our work through his comments and highlighting its strengths.
For more details please see the revised version manuscript.
Reviewer 2 Report
Authors prepare a methodological paper of audit and feedback intervention in emergency care for acute myocardial infarction in Italy. In this manuscript, authors measured 10 indicators to evaluate performance of each hospital, and they showed them in 16 hospitals. However, this article has not fully answered some of the questions due to insufficient description.
First, as authors mentioned as “The objectives of this work were to describe the A&F methodology for improving the quality of the in-hospital emergency care pathway for Acute Myocardial Infraction (AMI)” (L84), this manuscript may be a methodological paper, but the description of method seems to be poor. For example, regarding “reference value” authors used, it is not clear why authors defined so scientifically, and why authors did not include reference value in table 1, what method of adjustment of “demographics and clinical characteristics” (L214) is, how they used “reference value” in audit and feedback, and what details of “intervention for improving the quality of the in-hospital emergency care pathway” (L304) is. Authors may include some of description in supplemental files, but as methodological paper, authors should fully describe method of this study in the main manuscript.
Second, authors sometimes use the word “all”,(e.g., “All hospitals in the Lazio region have been invited to participate to a prospective observational study” (L899), and “A main Word document shows all the indicators for all the hospitals” (L122)), but it is not clear what “all” means (i.e., 50 or 16). Moreover, it is difficult to understand the description “invited hospitals’ (heath management and clinical specialists) (“feedback recipients”)” (L104). Authors clearly describe the manuscript.
Finally, authors assertively described some of sentences without citation as follows; “To reduce this variability and avoid suboptimal implementation of evidence-based practice, various strategies have been proposed, among them Audit and Feedback (A&F).” (L39), “A&F is commonly used to help health care providers to identify the gap between knowledge and practice and to improve quality of care. P” (L45), “Despite A&F as a quality improvement strategy is widely used, evidence suggest that the effects of such interventions vary greatly and are not improving over time.” (L49), “Hospitalizations for AMI have been progressively decreasing since 2015 while those for STEMI since 2010.” (L229), “when the feedback is delivered by a research team (and is this the case) this could be less effective than feedback delivered by a supervisor or a respected colleague.” (L352), and “In recent years, 30 days mortality from AMI has been significantly reduced. A” (L380), but it is difficult for readers to judge it without references as evidence for each description. Authors should add references for these descriptions.
Minor comments
L139. “TP” should be spelled out.
L263. Unit should be added for “≤50; 51-100; 101-150; >150”.
Extensive editing of English language may be required. For example, After the “words ”In Italy", you may need ”,".
Author Response
Dear Reviewer,
Response to Reviewer 2 Comments
Authors prepare a methodological paper of audit and feedback intervention in emergency care for acute myocardial infarction in Italy. In this manuscript, authors measured 10 indicators to evaluate performance of each hospital, and they showed them in 16 hospitals. However, this article has not fully answered some of the questions due to insufficient description.
Point 1: First, as authors mentioned as “The objectives of this work were to describe the A&F methodology for improving the quality of the in-hospital emergency care pathway for Acute Myocardial Infraction (AMI)” (L84), this manuscript may be a methodological paper, but the description of method seems to be poor. For example, regarding “reference value” authors used, it is not clear why authors defined so scientifically, and why authors did not include reference value in table 1, what method of adjustment of “demographics and clinical characteristics” (L214) is, how they used “reference value” in audit and feedback, and what details of “intervention for improving the quality of the in-hospital emergency care pathway” (L304) is. Authors may include some of description in supplemental files, but as methodological paper, authors should fully describe method of this study in the main manuscript.
Response Point 1:
We thank the Reviewer for this comment.
- We added more details to the description of the methodology (paragraph 2.1 and 2.2) and checked the references for any values used as comparator. We added a column to the Box1 (previously named Table 1) to better clarify the comparators used.
- Regarding the method of adjustment for demographics and clinical characteristics we added at line 237-240 of the revised manuscript the following sentence:
“The adjusted measures and related 95% CIs are calculated by generalized linear models with binomial distribution and logit as link function adjusting for demographics and clinical characteristics selected by means of a stepwise procedure (Supplementary material Table S3 and Table S4).”
Moreover in Supplementary material Table S3 and Table S4 we reported the link to the calculation protocol of each indicators with details about the demographics and clinical characteristics used for the adjustment.
Point 2: Second, authors sometimes use the word “all”,(e.g., “All hospitals in the Lazio region have been invited to participate to a prospective observational study” (L899), and “A main Word document shows all the indicators for all the hospitals” (L122)), but it is not clear what “all” means (i.e., 50 or 16). Moreover, it is difficult to understand the description “invited hospitals’ (heath management and clinical specialists) (“feedback recipients”)” (L104). Authors clearly describe the manuscript.
Response Point 2:
-At line 89-93 of the revised manuscript in the sentence: “All hospitals in the Lazio region were invited to participate to a prospective observational study aimed at evaluating the effectiveness of A&F intervention in promoting quality of care as regards AMI care pathway, centrally coordinated by DEP- Lazio.” was changed to:
“All fifty hospitals in the Lazio region were invited to participate to a prospective quasi-experimental, pre-post study with control group, aimed at evaluating the effectiveness of an A&F intervention in promoting quality of in-hospital emergency care for patients affected by AMI, centrally coordinated by DEP- Lazio.”.
-At line 142 of the revised manuscript we modified the sentence:
“A main Word document shows all the indicators for all the hospitals compared to standard values and regional means, and contain a brief textual description of results.”
as follows:
“A main document reports the results of the set of indicators calculated for all participating hospitals.”
As written in the results section, the hospitals joined to the intervention are sixteen.
-At line 102 of the revised manuscript we modified the sentence:
“At the beginning, on the 16th of December 2021, a kick off meeting was organized to present the project activities and to collect agreement to participate from representative professionals of invited hospitals’ (heath management and clinical specialists) (“feed-back recipients”). Before the intervention started, a questionnaire was administered to collect information about the state of implementation of A&F and assimilating activities in the participating hospital. Methods and results of this survey are described in the study by Angioletti et al [16].”
as follows:
“The experimental A&F intervention started in December 2021 and is scheduled to finish in September 2023. As the first step, a kick off meeting was organized to present the project activities and to collect agreement to participate from healthcare managers and clinicians - “feedback recipients”- of all fifty hospitals in the Lazio region. Before the intervention started, a questionnaire was administered to collect information about the state of implementation of A&F and assimilating activities in the participating hospital. Methods and results of this survey are described in the study by Angioletti et al [16].”
And move it in the section 2.2 A&F intervention, line 111-117.
Please see also Response 1 in which we report the changes made to improve the description of our work methodology.
Point 3: Finally, authors assertively described some of sentences without citation as follows; “To reduce this variability and avoid suboptimal implementation of evidence-based practice, various strategies have been proposed, among them Audit and Feedback (A&F).” (L39), “A&F is commonly used to help health care providers to identify the gap between knowledge and practice and to improve quality of care. P” (L45), “Despite A&F as a quality improvement strategy is widely used, evidence suggest that the effects of such interventions vary greatly and are not improving over time.” (L49), “Hospitalizations for AMI have been progressively decreasing since 2015 while those for STEMI since 2010.” (L229), “when the feedback is delivered by a research team (and is this the case) this could be less effective than feedback delivered by a supervisor or a respected colleague.” (L352), and “In recent years, 30 days mortality from AMI has been significantly reduced. A” (L380), but it is difficult for readers to judge it without references as evidence for each description. Authors should add references for these descriptions.
Response Point 3:
- At line 39 of the revised manuscript for the sentence “To reduce this variability and avoid suboptimal implementation of evidence-based practice, various strategies have been proposed, among them Audit and Feedback (A&F)” we added the references [2,3]:
- Flottorp, S. A.; Jamtvedt G.; Gibis B.; Mckee M. Using audit and feedback to health professionals to improve the quality and safety of health care. Belgian EU Presidency Conference on Investing in Europe's health workforce of tomorrow: scope for innovation and collaboration, La Hulpe, Belgium, 9-10 September 2010. Available online: https://apps.who.int/iris/handle/10665/332014?show=full.
- Davis DA, Mazmanian PE, Fordis M, Van Harrison R, Thorpe KE, Perrier L. Accuracy of physician self-assessment compared with observed measures of competence: a systematic review. JAMA. 2006, 296(9),1094–102. https://doi.org/10.1001/jama.296.9.1094.
- At line 46 of the revised manuscript for the sentence “Indeed, A&F is commonly used to help health care providers to identify the gap between knowledge and practice and to improve quality of care” we added the references [3,4]:
- Davis DA, Mazmanian PE, Fordis M, Van Harrison R, Thorpe KE, Perrier L. Accuracy of physician self-assessment compared with observed measures of competence: a systematic review. JAMA. 2006, 296(9),1094–102. https://doi.org/10.1001/jama.296.9.1094.
- Hysong S.J.; Kell H.J.; Petersen L.A.; Campbell B.A.; Trautner B.V. Theory-based and evidence-based design of audit and feedback programmes: examples from two clinical interventions studies. BMJ. Qual. Saf. 2017, 26,323-334. https://doi.org/10.1136/bmjqs-2015-004796.
- At line 51 of the revised manuscript for the sentence “Despite A&F as a quality improvement strategy is widely used, evidence suggest that the effects of such interventions vary greatly and are not improving over time” we added the references [5,6]:
- Ivers N.; Jamtvedt G.; Flottorp S.; Young J.M.; Odgaard-Jensen J.; French S.D.; O'Brien M.A.; Johansen M.; Grimshaw J.; Oxman A.D. Audit and feedback: effects on professional practice and healthcare outcomes. Cochrane Database of Sys-tematic Reviews. 2012, 6, 1465-1858. https://doi.org/10.1002/14651858.CD000259.pub3.
- Ivers N.M.; Grimshaw J.M.; Jamtvedt G.; Flottorp S.; O’Brien M.A.; French S.D.; Young J.; Odgaard-Jensen J. Growing lit-erature, stagnant Science? Systematic review, meta-regression and cumulative analysis of audit and feedback interventions in health care. J. Gen. Intern. Med. 2014,29,1534-41. https://doi.org/ 10.1007/s11606-014-2913-y.
- At line 256 of the revised manuscript for the sentence “Hospitalizations for AMI have been progressively decreasing since 2015 while those for STEMI since 2010.” we added the references [24]:
- P.Re.Val.E edition 2022 web site. Volume of hospitalization for AMI from 2012 to 2021. Available online: https://www.dep.lazio.it/prevale2022/risultati/tipo5/home_tipo5.php?ind=122&tipo=5&area=1 (accessed on 15 September 2022).
- At line 385 of the revised manuscript we changed the sentence:
“On the other hand, we need to consider that when the feedback is delivered by a research team (and is this the case) this could be less effective than feedback delivered by a supervisor or a respected colleague.”
to
“On the other hand, it must be considered that feedback delivered by a research team, as in our case, might be less effective than that delivered by a supervisor or a respected colleague [9].”
And we added the references [9]:
- Brehaut J.C.; Colquhoun H.L.; Eva K.W.; Carroll K.; Sales A.; Michie S.; Ivers N.; Grimshaw J.M. Practice feedback inter-ventions: 15 suggestions for optimizing effectiveness. Ann Intern Med. 2016, 164 (6), 435-41. https://doi.org/10.7326/M15-2248.
- At line 416 of the revised manuscript for the sentence “In recent years, 30 days mortality from AMI has been significantly reduced.” we added the references [27]:
- Laforgia P.L.; Auguadro C.; Bronzato S.; Durante A. The Reduction of Mortality in Acute Myocardial Infarction: From Bed Rest to Future Directions. Int J Prev Med. 2022 Apr 8;13:56. https://doi: 10.4103/ijpvm.IJPVM_122_20.
Point 4: Minor comments
L139. “TP” should be spelled out.
L263. Unit should be added for “≤50; 51-100; 101-150; >150”.
Response Point 4: At line 93 of the revised manuscript we, after english revision, we modified “Team Project” with “project team (PT)”
At line 294 and 295 of the rivised manuscript we added the unit for “≤50; 51-100; 101-150; >150” as “≤50; 51-100; 101-150; >150 number of PTCA”.
Point 5: Extensive editing of English language may be required. For example, After the “words ”In Italy", you may need ”,".
Response Point 5: As suggested, we carried out a careful linguistic review of the work.
For more details please see the revised version manuscript.
Reviewer 3 Report
Dear Authors
It is an honor to be invited to review this article which named “EASY-NET Program: Methods and Preliminary Results of an Audit & Feedback Intervention in the Emergency Care for Acute Myocardial Infarction in the Lazio Region, Italy.”
The EASY-NET Network Program (NET-2016-02364191), specifically Work Package 1 - Lazio, is conducting a study to determine the effectiveness of a structured Audit & Feedback (A&F) intervention compared to a regional periodic publication of indicators delivered online, in improving emergency healthcare for acute myocardial infarction (AMI) in terms of appropriateness and timeliness. The A&F intervention involves sending regular email reports to healthcare managers and clinicians in each participating hospital. These reports contain volume and quality indicators (both process and outcome) calculated from the Health Information System of the Lazio Region, which are then compared to regional averages, target values, and values from hospitals with similar activity levels. Recipients are encouraged to organize audit meetings to identify potential issues in care pathways and develop improvement plans where necessary. Sixteen facilities are participating, with twelve having high activity levels for all volume indicators, and three with low activity levels. Quality indicators show that four facilities have no critical indicators, three have no critical indicators but have average results in at least one indicator, and six have at least one critical value. The initial report identified issues across various indicators in multiple facilities. The audit meetings will address these issues, and improvement actions will be defined and monitored through subsequent reporting to achieve continuous quality improvement.
From the display of this article, we recognized the Audit & Feedback (A&F) interventions are effective strategies for improving healthcare quality. With the development of information technology and the improvement of related measures, I also believe that EASY-NET program will become an effective measure to improve the quality of medical care in the region.
The implementation of this network may have some shortcomings, such as insufficient coordination among different hospitals. Additionally, such a network may potentially affect the efficiency of diagnosis and treatment.
Sincerely,
There are no obvious grammar and writing mistakes as far as I am concerned.
Author Response
Dear Reviewer,
Response to Reviewer 3 Comments
Dear Authors
It is an honor to be invited to review this article which named “EASY-NET Program: Methods and Preliminary Results of an Audit & Feedback Intervention in the Emergency Care for Acute Myocardial Infarction in the Lazio Region, Italy.”
The EASY-NET Network Program (NET-2016-02364191), specifically Work Package 1 - Lazio, is conducting a study to determine the effectiveness of a structured Audit & Feedback (A&F) intervention compared to a regional periodic publication of indicators delivered online, in improving emergency healthcare for acute myocardial infarction (AMI) in terms of appropriateness and timeliness. The A&F intervention involves sending regular email reports to healthcare managers and clinicians in each participating hospital. These reports contain volume and quality indicators (both process and outcome) calculated from the Health Information System of the Lazio Region, which are then compared to regional averages, target values, and values from hospitals with similar activity levels. Recipients are encouraged to organize audit meetings to identify potential issues in care pathways and develop improvement plans where necessary. Sixteen facilities are participating, with twelve having high activity levels for all volume indicators, and three with low activity levels. Quality indicators show that four facilities have no critical indicators, three have no critical indicators but have average results in at least one indicator, and six have at least one critical value. The initial report identified issues across various indicators in multiple facilities. The audit meetings will address these issues, and improvement actions will be defined and monitored through subsequent reporting to achieve continuous quality improvement.
From the display of this article, we recognized the Audit & Feedback (A&F) interventions are effective strategies for improving healthcare quality. With the development of information technology and the improvement of related measures, I also believe that EASY-NET program will become an effective measure to improve the quality of medical care in the region.
Point 1: The implementation of this network may have some shortcomings, such as insufficient coordination among different hospitals. Additionally, such a network may potentially affect the efficiency of diagnosis and treatment.
There are no obvious grammar and writing mistakes as far as I am concerned.
Response Point 1: First of all, we thank the Reviewer for his favourable comments and considerations regarding our work and its possible positive impact in improving the quality of healthcare in the Lazio Region.
Concerning possible shortcomings in the implementation of this network, giving hospitals the possibility to organize the audit meetings independently can certainly create problems of homogeneity and coordination among the participating structures but, on the other hands, it gives them the possibility of adapting the meetings to their own needs. We added this consideration in the discussion of the revised manuscript (line 450):
“Moreover, the research team produces and sends well defined and standardized feed-back reports thus giving hospitals the possibility to organize the audit meetings inde-pendently. This can certainly create problems of heterogeneity and coordination among the participating facilities but gives them the possibility of adapting the meetings to their own needs. In any case, to improve coordination, periodic meetings for discussion were organized and numerous contacts were held via email and telephone with all the participants.”
For more details please see the revised version manuscript.
Reviewer 4 Report
I congratulate the authors on a very interesting manuscript describing the methodology for the methodology of the audit and feedback of the EASY-NET program (which aims to improve quality of the emergency care pathway for AMI) across 16 faculties in Lazio, Italy.
After introducing the Delphi method in line 139, can you provide a sentence to summarise what it is?
Please provide a reference for the Decreto Ministerial 70/2015
In table 2 -4 why is there data missing for facilities 12-16?
Well done on an this methodology, I look forward to reading the results following completion of the intervention.
Please review the grammer and spelling for the manuscript, below are a few suggestions:
Line 85 - “infarction” spelt incorrectly.
Line 221 - tense.
Line 352 - grammar.
Line 387 - Only need to provide the definition of the acronym for ER the first time it is used in the text.
Author Response
Dear Reviewer,
Response to Reviewer 4 Comments
I congratulate the authors on a very interesting manuscript describing the methodology for the methodology of the audit and feedback of the EASY-NET program (which aims to improve quality of the emergency care pathway for AMI) across 16 faculties in Lazio, Italy.
Point 1: After introducing the Delphi method in line 139, can you provide a sentence to summarise what it is?
Response Point 1: As suggetsed we added at line 162-163 of the revised manuscript the following sentense “a group process used to survey and collect the opinions of experts on a particular subject”
Point 2: Please provide a reference for the Decreto Ministerial 70/2015
Response Point 2: We added the following reference at line 180 of the revised manuscript.
- Ministerial decree 2 April 2015 n. 70. (G.U. 4 2015, n. 127). Available online: https://www.camera.it/temiap/2016/09/23/OCD177-2353.pdf (Accessed on 20 September 2022).
Point 3: In table 2 -4 why is there data missing for facilities 12-16?
Response Point 3: We thank the reviewer for these suggestions that help make the results shown more understandable.
In Table 2 there are some facilities (Facility 13,14,15 and 16) that have not recorded hospitalisations of patients with STEMI treated with PTCA in the considered years. For greater clarity we have replaced the underscore in Table 2 with zero.
In Tables 3 and 4 there are facilities for which the adjusted proportion could not be estimated because the sample size was too small (as written at line 308).
In this regard, we have modified the following sentence in line 308-309 of the revised manuscript:
“For three facilities it is not possible to calculate the adjusted mortality and the crude proportion is reported, which is higher than the regional value in all cases.”
as follows:
“Three facilities had very low values and for them is not possible to calculate the adjusted mortality. The crude proportion is reported and is higher than the regional value in all cases.”
Well done on an this methodology, I look forward to reading the results following completion of the intervention.
Point 4: Please review the grammer and spelling for the manuscript, below are a few suggestions:
Line 85 - “infarction” spelt incorrectly.
Line 221 - tense.
Line 352 - grammar.
Line 387 - Only need to provide the definition of the acronym for ER the first time it is used in the text.
Response Point 4: As suggested, we carried out a careful linguistic review of the work.
For example, as suggested, we made the following corrections:
- we have replaced “infraction with “infarction”.
- we modify the sentence at line 246 of the revised manuscript according to the reviwer comment and, in addition we corrected the number of facilities in the cardiological emergency network in Lazio region as follows:
“A total of 16 out of 50 facilities of the Lazio region participates to the project for the AMI pathway evaluation.”
becomes:
“Out of the total of 43 facilities in the cardiological emergency network in Lazio region, 16 participate in the intervention”.
- At line 385 of the revised manuscript we changed the sentence:
“On the other hand, we need to consider that when the feedback is delivered by a research team (and is this the case) this could be less effective than feedback delivered by a supervisor or a respected colleague.”
to
“On the other hand, it must be considered that feedback delivered by a research team, as in our case, might be less effective than that delivered by a supervisor or a respected colleague.”
- the acronym for Emergency Room (ER) is reported for the first time at line 170 in the footnotes of Table 1 and in the text at line 422 of the revised manuscript. At line 424 of the revised manuscript we reported only the acronym ER previously introduced.
For more details please see the revised version manuscript.
Round 2
Reviewer 2 Report
There may be some of editorial issues (e.g., space in L129 and period in L364), but I have no further major comments.
There may be some of editorial issues (e.g., space in L129 and period in L364).
Author Response
We thank the reviewer for editorial suggestions and have removed the spaces at lines 129 and 364.